# Evaluation of Objective and Subjective Swallowing Outcomes in Patients with Dysphagia Treated for Head and Neck Cancer

**DOI:** 10.3390/jcm11030692

**Published:** 2022-01-28

**Authors:** Hsin-Hao Liou, Shu-Wei Tsai, Miyuki Hsing-Chun Hsieh, Yi-Jen Chen, Jenn-Ren Hsiao, Cheng-Chih Huang, Chun-Yen Ou, Chan-Chi Chang, Wei-Ting Lee, Sen-Tien Tsai, David Shang-Yu Hung

**Affiliations:** 1Department of Otolaryngology, National Cheng Kung University Hospital, College of Medicine, National Cheng Kung University, Tainan 704, Taiwan; dennis70011@gmail.com (H.-H.L.); tsaisuwei@gmail.com (S.-W.T.); em75311@email.ncku.edu.tw (Y.-J.C.); hsiaojr@mail.ncku.edu.tw (J.-R.H.); cs841063@yahoo.com.tw (C.-C.H.); ojy1@mail.ncku.edu.tw (C.-Y.O.); guitarist217@yahoo.com.tw (C.-C.C.); wendelllee92@yahoo.com.tw (W.-T.L.); t602511@mail.ncku.edu.tw (S.-T.T.); 2Institute of Clinical Pharmacy and Pharmaceutical Sciences, College of Medicine, National Cheng Kung University, Tainan 704, Taiwan; s1234567y@gmail.com

**Keywords:** head and neck cancer, dysphagia, swallowing, subjective assessment, objective assessment, outcomes

## Abstract

We evaluated objective and subjective swallowing function outcomes in patients with dysphagia treated for head and neck cancer (HNC) and identified risk factors for poor swallowing outcomes. Patients undergoing videofluoroscopic swallowing studies (VFSS) between January 2016 and March 2021 were divided into four groups according to primary tumor sites; post-treatment dysphagia was assessed. The penetration–aspiration scale (PAS) and bolus residue scale (BRS) were used to objectively assess swallowing function through VFSS. The Functional Oral Intake Scale (FOIS) was used for subjective analyses of swallowing statuses. To account for potential confounding, important covariates were adjusted for in logistic regression models. Oropharyngeal tumors were significantly more likely to have poor PAS and BRS scores than oral cavity tumors, and the patients with nasopharyngeal tumors were significantly less likely to have poor FOIS scores. Old age, having multiple HNCs, and a history of radiotherapy were associated with an increased odds of poor PAS scores (for all types of swallows), poor BRS scores (for semiliquid and solid swallows), and poor FOIS scores, respectively. This indicates using only subjective assessments may not allow for accurate evaluations of swallowing function in patients treated for HNC. Using both objective and subjective assessments may allow for comprehensive evaluations.

## 1. Introduction

Survival of patients with head and neck cancer (HNC) has improved considerably in recent decades due to improved public health awareness and the use of management strategies, such as radiotherapy or surgery [1,2]. With an increase in the number of long-term survivors of HNC, there has been an increase in the amount of attention paid to and emphasis placed on the sequelae of HNCs and adverse effects of managing them. Dysphagia involves difficulty in swallowing, and it affects 60–75% of patients treated for HNC [3]; it is an issue that may affect one’s quality of life, nutrition status, and long-term survival [4]. Dysphagia could be caused by the presence and/or treatment of a tumor [5,6], and it involves the retention of food boluses in the oropharynx or hypopharynx; such retention may result in the spillage of food into the airway, leading to aspiration [7].

Predictive factors of dysphagia in patients with HNC have been previously studied in several studies. In such studies, it was found that the receipt of radiation therapy (including radiation field and dose) [8,9], tumor stage [9,10,11], the presence of preexisting dysphagia (which is a risk factor for the development of long-term dysphagia) and underlying diseases (such as stroke) [11], patient age [12,13], region of residence, and having a history of significant alcohol consumption were risk factors for the development of dysphagia [10]. Different primary tumor sites have also been found to be associated with different odds for the development of dysphagia. Aylward et al. found that patients with hypopharyngeal cancer were associated with an increased risk of developing long-term dysphagia, and Xu et al. found that presence of hypopharyngeal tumors was a risk factor for the development of aspiration pneumonia [11,13].

Multiple instruments have been used to assess patients’ swallowing function and evaluate the severity and various aspects of dysphagia. In most past studies involving evaluations of swallowing impairment, videofluoroscopic swallowing studies (VFSSs) have been used [14,15,16]. Among the different means used to evaluate airway invasion during swallowing, the penetration–aspiration scale (PAS) [15,17,18,19] is the most widely used scale. Although the use of the bolus residue scale (BRS) has not been reported as much as the use of the PAS, the BRS is another scale that can be applied (with high specificity) to evaluate bolus residue observed using VFSSs [20]. Using the PAS, the risk of choking or aspiration can be clearly defined, and using the BRS, the rate and location of pharyngeal retention can be easily and reliably evaluated. In order to evaluate swallowing function, in addition to approaches involving the use of VFSSs, the Functional Oral Intake Scale (FOIS) [21] has also been used to document patients’ functional levels of oral intake [14,22]. However, a correlation has not been found between PAS scores and FOIS scores in past studies, which suggests that there are differences between the findings of objective assessments (such as assessments involving the use of the PAS) and functional measurements (such as FOIS scores) [23,24]. For comprehensive evaluations of dysphagia in patients, it may be necessary to use multiple methods simultaneously.

However, only a few past studies have involved the use of both objective and subjective assessment methods to evaluate swallowing outcomes in patients with HNC, especially with respect to patients who have already undergone cancer treatment, which is considered a risk factor for the development of dysphagia. Thus, in the present study, we aimed to describe swallowing outcomes in patients with HNC through subjective and objective assessments and identify risk factors for the worsening of dysphagia.

## 2. Methods

### 2.1. Design and Study Population

In this retrospective, cross-sectional, observational study, patients treated for HNC who had visited the otorhinolaryngology outpatient clinic of National Cheng Kung University Hospital (NCKUH) with a complaint of difficulty swallowing and undergone VFSSs between January 2016 and March 2021 were included. A patient was included in the study population if the patient had been diagnosed with HNC, had already received operative treatment, radiotherapy, or both, and had VFSS results of appropriate quality, including accurate observations of pharyngeal structures. The operative treatments in these patients with HNC included partial laryngectomy, glossectomy, pharyngectomy, and local or free flap reconstruction. To simplify analyses, for each patient, only the first VFSS recording made after the patient had been treated using surgery, or radiotherapy was considered. Patients with underlying conditions such as stroke, dementia or neuromuscular diseases that had already caused or may cause dysphagia were excluded. Furthermore, patients aged <18 years or >75 years, patients for whom the use of VFSSs was contraindicated, patients who had received treatment for dysphagia before they underwent VFSSs, patients diagnosed with cancer that had not originated in the head or neck, and patients who had laryngeal cancer and/or had undergone total laryngectomy (which leads to the elimination of aspiration) were excluded.

The included patients were divided into the following four groups based on the primary site of HNC: the oral cavity, nasopharynx, oropharynx, and hypopharynx groups. HNC was staged according to the 8th edition of the American Joint Committee on Cancer’s tumor, node, and metastasis classification system and staging criteria [25]. None of the patients were undergoing radiotherapy when their swallowing function was evaluated, and none of them had undergone head and neck surgery during the month preceding the time they underwent VFSS evaluation [26]. The outcome of interest was the included patients’ swallowing function. The PAS, BRS, and FOIS were the three scoring scales used to assess swallowing function. For objective assessments of swallowing function, the PAS and BRS were used to evaluate observations made during VFSSs, whereas for subjective assessments of the patients’ swallowing function, the FOIS was used.

### 2.2. Objective and Subjective Analyses of Swallowing Function

The included patients’ swallowing outcomes were objectively assessed through evaluations of observations made using VFSSs; in this regard, aspiration and post-swallow residue were assessed. Aspiration was assessed using the PAS, which is an eight-point scale, with scores ranging from 1 (which indicates that material does not enter the airway of an individual) to 8 (which indicates that material enters the airway of an individual, passes below the vocal folds, and no effort is made to eject the material) [17]. To determine PAS scores, lateral views were used. We considered that high PAS scores indicated poor swallowing function, and a PAS score > 2 indicated that a patient had symptomatic dysphagia and clinically abnormal swallowing function [19]. Post-swallow residues were assessed using the BRS, which is used to score the presence of post-swallow residue in the epiglottic vallecula, piriform sinuses, and/or posterior pharyngeal wall. If no residue was present in any of these structures, the patient was given a score of 1; conversely, if residue was present in each of these three structures, the patient was given a score of 6. If an individual has a high BRS score, it indicates that, after swallowing, a relatively large amount of post-swallow food residue is present in the individual, and the risk of the occurrence of aspiration may be high for patients with high BRS scores [20]. BRS scores were double checked by observing both lateral and frontal anterior views. If the BRS score for a patient was >3, we considered that the patient had a clinically significant amount of post-swallow residue [20].

The FOIS was used to subjectively assess swallowing function. If the FOIS score for an individual is high, the individual’s swallowing function is considered to be good [21]. a score of 7 is given to patients whose oral intake ability is not restricted. In contrast, a score of 1 is given to patients who cannot ingest food orally; such patients either need to undergo gastrostomy or need to be fed exclusively through tube feedings. If a patient has an FOIS score < 4, the score represents that, for the maintenance of the patient’s daily nutrition intake, the patient is either fully or partially dependent on tube feeding.

### 2.3. Assessment of Swallowing Function in NCKUH

To all included patients, VFSS was performed by a speech pathologist with more than 10 years of experience. Standardized VFSS was performed using a remote-controlled fluoroscope (AXIOM Luminos dRF, Siemens, Munich, Germany). The images were recorded with a frame rate of 30 frames per second, through which dynamic images derived in real-time and static images for different frames could be displayed. The patients were asked to sit on a chair in a manner that allowed for the obtainment of images in both lateral and anterior–posterior views. Each patient swallowed three portions of a standard formula (a 5 mL liquid bolus, 10 mL semiliquid bolus, and solid bolus (a biscuit of size 3 cm × 3 cm × 1 cm)) with a spoonful of barium sulfate (Baritop LV, Kaigen Pharma Co Ltd., Osaka, Japan) added. The liquid bolus and the semiliquid bolus had thin and mildly thick consistencies, respectively, according to the International Dysphagia Diet Standardization Initiative framework. Three types of barium-containing boluses were used because aspiration or the presence of post-swallow residue caused by dysphagia may be affected by bolus textures [27]. For each patient, we recorded the VFSS video until the patient completed the entire swallowing process, which meant that the boluses had entered the patient’s esophagus. An experienced radiologist and a licensed speech pathologist together analyzed the VFSS images of the included patients.

### 2.4. Other Covariates

In addition to collecting information on the type of HNC in each patient, the stage of cancer, and swallowing scores, we also collected information regarding the patients’ sex and age at the time of VFSS assessment and their medical history with respect to surgery and radiotherapy. Furthermore, we determined whether the patients had been diagnosed with multiple HNCs, including recurrent cancer of the original or a second primary HNC at a different site.

### 2.5. Statistical Analyses

Characteristics of the study population have been presented using descriptive statistics. Mean and standard deviation values or median values and interquartile ranges have been used to present continuous variables; frequencies and percentages have been used to present categorical variables. To evaluate differences among the characteristics of patients with different types of HNC, analysis of variance or the Kruskal–Wallis test was used to analyze continuous variables, and the chi-square test or Fisher’s exact test was used to analyze categorical variables. To investigate the association between different tumor types and PAS, BRS, and FOIS scores, ordinal logistic regression, in which swallowing scores were treated as ordinals instead of continuous outcomes, was used. We considered that high scores represented poor outcomes. In order to conduct ordinal logistic regression in a consistent manner, we recoded FOIS scores, with an FOIS score of 1 being recoded as a score of 7, and a score of 7 being recoded as a score of 1. To adjust for confounders, we accounted for patient demographics, tumor T stages, whether a patient had undergone surgery or radiotherapy, and whether or not a patient had multiple HNCs. Furthermore, in order to identify patients with clinically abnormal swallowing function, we classified the patients according to their VFSS scores, and if a patient had a PAS score ≥ 3 or a BRS score ≥ 4, the patient was considered to have clinically abnormal swallowing function. We calculated the proportion of patients in each group who had abnormal PAS and BRS scores for liquid, semiliquid, and solid swallows. Thereafter, we applied multivariate logistic regression to compare the risk of having abnormal swallowing outcomes for patients with different tumor types; for this purpose, the oral cavity group was considered the reference group. The same covariates considered in linear regression were adjusted for in multivariate logistic regression. Variance inflation factors were calculated to assess if collinearity existed between any variables, and if the value of the variance inflation factor was <10, it was considered that there was no significant collinearity. Odds ratios (ORs), 95% confidence intervals (CIs), and associated *p* values were determined for different tumor types and other covariates. *p* values < 0.05 were considered statistically significant. All analyses were performed using the SAS 9.4 software (SAS Institute, Cary, NC, USA).

## 3. Results

### 3.1. Patient Characteristics

During the 5-year study period, 172 patients who had already received treatment for HNC were referred to a speech therapist for the assessment of dysphagia through VFSSs. Of these patients, four were excluded because they had clinically manifested dysphagia caused by stroke prior to being diagnosed with HNC, and nine were excluded because they had laryngeal cancer and/or had undergone total laryngectomy (Figure 1). After VFSS recordings had been reviewed, six more patients were excluded because clear VFSS images were not available for those patients. Ultimately, a total of 153 patients were included in the study.

With respect to the classification of patients based on primary tumor sites, among the included patients, 105 (69%) patients were included in the oral cavity group, 14 (9%) patients were included in the nasopharynx group, 17 (11%) patients were included in the oropharynx group, and 17 (11%) patients were included in the hypopharynx group. In total, the mean age of the patients included in the study was 55.05 years (standard deviation: 9.49 years), and more than 90% of patients were men. The median duration between the time the patients underwent surgery to the time they underwent VFSSs was 9.75 months; the median duration between the time the patients underwent radiotherapy to the time they underwent VFSSs was 10.9 months. Approximately half of the patients underwent both surgery and radiotherapy for the treatment of HNC. Among the included patients, 50–60% of the patients had HNC of stage 3 or 4 at the time of diagnosis, and 30% of the patients had previously had HNC in a different region (in this study, such patients have been described as patients who had multiple HNCs). The four groups of patients had similar characteristics; however, compared to the other groups, the nasopharynx group had a smaller proportion of men and patients who had undergone surgery. All the patients in the nasopharynx group had undergone radiotherapy, and none of the patients had multiple HNCs. Detailed demographic characteristics of the patients have been presented in Table 1.

### 3.2. Tumor Types and Swallowing Outcomes

In total, the median PAS and BRS scores for all the included patients were 1 and 3, respectively, for all types of swallows (liquid, semiliquid, or solid bolus swallows) (Table 2). The median FOIS score was 4.5. When the scores for the different groups of patients were compared, we found that, compared to the PAS scores for solid, semiliquid, and liquid swallows for the patients in the oral cavity group (median score: 1), the PAS scores for the patients in the oropharynx group were significantly higher (median scores: 3–4). With respect to BRS scores, no significant differences were found among the four groups of patients. With respect to FOIS scores, significant differences were observed among the four groups. The nasopharynx group had the highest median score (6) when compared to other groups of patients. Compared to the oral cavity group, the oropharynx and hypopharynx groups had significantly higher odds of having high PAS scores for liquid swallows (ORs: 2.8 (*p* = 0.043) and 3 (*p* = 0.030) for the oropharynx and hypopharynx groups, respectively) (Table 3). It was noted that, in all types of swallowing, oropharynx group manifested consistent ORs at approximately 3 when compared to the oral cavity group. With respect the risk of having high PAS scores, the OR associated with a 1-year increase in age was found to be 1.04 (*p* values for different types of swallows ranged from 0.031 to 0.042); with respect to the risk of having high BRS scores for semiliquid and solid swallows, the OR associated with having multiple HNCs was found to be 2.2 (*p* values for different types of swallows ranged from 0.023 to 0.036). With respect to FOIS scores, compared to patients with oral cavity group, the patients who had nasopharyngeal tumors were found to be significantly less likely to have poor FOIS scores (*p* = 0.025). However, with respect to the risk of having poor FOIS scores, the OR associated with the receipt of radiotherapy was found to be 5.1 (*p* = 0.016) (Table 4).

Approximately 30% and 40% of the patients had abnormal PAS and BRS scores, respectively (Table 5). With respect to PAS scores, compared to the other groups, the oral cavity group had a relatively smaller proportion of patients with abnormal swallowing outcomes. With respect to BRS scores, no significant difference was found among the four groups of patients. The results of multivariate logistic regression indicated that, with respect to the risk of having poor PAS scores for solid, liquid, and semiliquid swallows, compared to the oral cavity group, the oropharynx group had significantly higher ORs (*p* values: 0.032–0.036) (Table 6). With respect to the risk of having abnormal PAS scores for liquid swallows, age was found to be associated with a high OR (OR: 1.06, 95% CI: 1.01–1.11). With respect to the risk of having abnormal BRS scores for solid swallows, the OR associated with having multiple HNCs was found to be 2.24 (95% CI: 1.08–5.42).

## 4. Discussion

By utilizing both objective and subjective assessment methods, we found that among patients who had been treated for HNC and had already developed dysphagia, patients who had oropharyngeal tumors had an increased odds of having poor swallowing outcomes during VFSSs. Furthermore, with respect to subjective assessments, we found that, compared to patients who had tumors in the oral cavity, patients who had nasopharyngeal tumors had better oral food intake. In this study, we also identified predictive factors for the worsening of dysphagia; in this regard, increased age was found to be associated with poor PAS scores, having multiple HNCs was found to be associated with poor BRS scores, and the receipt of radiotherapy was found to be associated with poor FOIS scores.

We also found that, compared to patients who had oral cavity tumors, the odds of aspiration with solid, liquid, and semiliquid swallows was higher for patients who had oropharyngeal tumors; this finding is similar to the findings of previous studies. It has been mentioned in past studies that patients with advanced-stage oropharyngeal and hypopharyngeal tumors and those with cervical esophageal cancer have severe dysphagia [5,11,28]. One study reported that, compared to patients with laryngeal tumors, patients with oropharyngeal tumors have significantly worse activity limitation with respect to semisolid swallows; activity is particularly limited 3 months after the receipt of treatment, and it improves 6 months after treatment, with only a transient risk of airway penetration [16,29]. Considering the physiological aspects of oropharyngeal events of swallowing, after a patient with HNC has been treated using either surgery or radiotherapy, changes in the lingual driving force, pharyngeal clearing forces, hypopharyngeal suction pump, and laryngeal competence may affect the pharyngeal phase of swallowing in that patient [30].

Interestingly, in this study, we found that patients who had hypopharyngeal tumors bear a higher odd of aspiration or penetration with liquid swallows compared to oral cavity group. In order to swallow a liquid bolus, the tongue acts as a reservoir to hold the bolus; the linguavelar valve is tightly closed until the oral transit phase begins and the bolus is pushed to the posterior oral cavity, which is when the pharyngeal phase of swallowing begins. For the ingestion of a solid bolus, as the bolus is masticated, it aggregates primarily in the vallecula, but can also reach the piriform sinuses [31,32]. In most surgeries for the treatment of oropharyngeal or hypopharyngeal cancer, important structures involved in the act of swallowing may be resected, and the effect of such surgeries on associated muscles and nerves may cause sensory–motor dysfunction and swallowing incoordination. On the contrary, radiotherapy may lead to pathophysiological changes, such as the development of lymphedema, fibrosis, and adhesion or synechia formation in the oropharyngeal and laryngeal mucosal membrane, pharyngeal constrictor muscles, suprahyoid musculature, or “mylohyoid–geniohyoid complex” [33,34]. Additionally, radiotherapy can cause sensory impairment in the tracheal area, which may lead to the occurrence of silent aspiration [7]. For these reasons, cancer treatment may cause a person to lose their ability to establish contact between the posterior portion of the tongue and the soft palate, leading to a separation between the oral and pharyngeal phases of swallowing. An early arrival of a liquid bolus in the hypopharynx can cause the bolus to penetrate the trachea before the closure of the larynx.

In this study, we noticed that the odds of aspiration increased with age; this association between the risk of aspiration and age has been demonstrated in many previous studies. Shune et al. had reported the association between the severity of dysphagia and age [35]. In another study, it was found that, among patients who receive radiotherapy, increased age may be associated with the occurrence of late aspiration [36]. Xu et al. also demonstrated that nearly 25% of elderly patients develop aspiration pneumonia within 5 years of receiving chemoradiotherapy for the treatment of HNC [13]. In the elderly, there are multiple causes of aspiration, such as the loss of protective laryngeal reflexes, loss of central coordination (which causes neurogenic dysphagia), and significant reductions in the cross-sectional area of the sphincter opening [37].

In contrast to the PAS, which was used as a tool to objectively evaluate swallowing safety, the BRS was used to evaluate swallowing efficiency. To the best of our knowledge, this is the first study in which the BRS was used to evaluate swallowing outcomes in patients treated for HNC. We found that, compared to other patients, the patients who had oropharyngeal tumors had higher BRS scores for all types of swallows (solid, liquid, and semiliquid) and the patients who had multiple HNCs of the head and neck had higher BRS scores for semiliquid and solid swallows. The application of the BRS allows for qualitative assessments, and it is an easy-to-use method for either expert or non-experts [20,38]. Patients who had oropharyngeal tumors were found to have high PAS and BRS scores, indicating that, with respect to both swallowing safety and efficiency, such patients had poor swallowing outcomes. With respect to the patients who had multiple HNCs, we assumed that the pathophysiological changes caused by multiple tumors led to decreased swallowing efficiency in these patients.

With respect to subjectively assessed swallowing outcomes, we found that patients with nasopharyngeal tumors had better FOIS scores. Radiotherapy and concurrent chemoradiation therapy were the main treatment methods used to treat nasopharyngeal tumors; the fact that among patients with nasopharyngeal tumors, the application of radiation in radiotherapy is confined to affected areas may explain why compared to swallowing dysfunction in patients who had oral cavity, hypopharyngeal, or oropharyngeal tumors that in patients who had nasopharyngeal tumors is minor.

To the best of our knowledge, this is one of the first studies to combine PAS, BRS, and FOIS to evaluate swallowing outcomes in patients treated for HNC. A combined use of multiple measures may help healthcare professionals to evaluate patients’ swallowing function more comprehensively and hence aid in early identification of dysphagia [24]. Dysphagia may result in aspiration, penetration, dehydration, malnutrition, and increase other morbidities in patients with HNC [39]. These would further decrease patients’ nutritional function and quality of life [40]. Our study implied that early identification of the dysphagia could be clinically significant, as it could facilitate early intervention or management to prevent further complications. However, this study has certain limitations. The small and unequal number of patients in the different patient groups may have affected the statistical power, and the scores for the patients were rarely normally distributed. However, we used ordinal logistic regression to account for data characteristics and reach appropriate conclusions. Another limitation is that we only included patients who had been treated for HNC and had already developed dysphagia; thus, the results may not be generalizable to other groups of patients with HNC.

## 5. Conclusions

Using an objective assessment tool, we found that the risk of aspiration and having bolus residue was high for patients with oropharyngeal tumors. In contrast, patients with nasopharyngeal tumors revealed better swallowing function using subjective assessment. These findings indicated that, compared to exclusively using only a single tool, complementary application of both objective and subjective assessments may improve evaluations of dysphagia in patients with HNC. We also found that increased age, having multiple HNCs, and receiving radiotherapy were predictive factors for poor PAS scores, BRS scores, and FOIS scores, respectively. Clinicians should take these risk factors into account when they assess swallowing function and design rehabilitation programs for these patients.

## Figures and Tables

**Figure 1 jcm-11-00692-f001:**
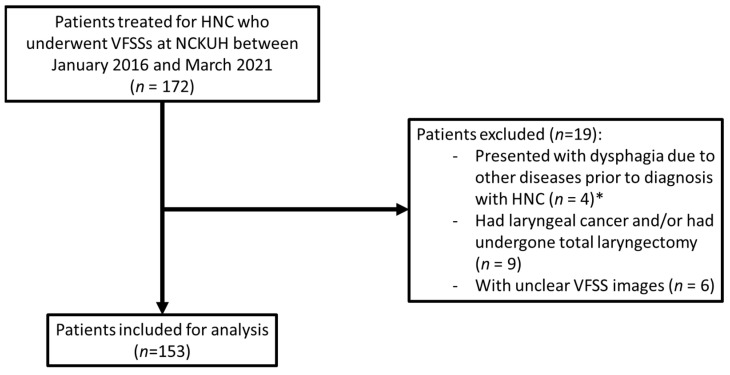
Flowchart depicting the inclusion of patients into the study population; * All 4 patients presented with dysphagia due to stroke before HNC diagnosis. Abbreviations: HNC, head and neck cancer; NCKUH, National Cheng Kung University Hospital; VFSSs, videofluoroscopic swallowing studies.

**Table 1 jcm-11-00692-t001:** Characteristics of included patients.

			Different Tumor Types:								
	Total		Oral Cavity		NPC		Oropharynx		Hypopharynx		
	*n* = 153		*n* = 105		*n* = 14		*n* = 17		*n* = 17		
	*n*	(%)	*n*	(%)	*n*	(%)	*n*	(%)	*n*	(%)	*p* value
Male	143	(93)	101	(96)	9	(64)	16	(94)	17	(100)	<0.001
Age (years), mean ± SD	55.05	±9.49	54.93	±10.11	53.07	±6.89	57.71	±9.51	54.71	±7.14	0.581
Time from HNC Dx *, median (IQR)	12	(26.4)	10.8	(24)	23.4	(43.2)	12	(31.2)	16.8	(14.4)	0.131
Time from surgery *, median (IQR)	9.75	17.45	9.25	20.9	14.9	0	10.65	24	10.2	7.3	0.884
Time from radiotherapy *, median (IQR)	10.9	23.45	7.25	29.5	24.3	54.8	7.8	12.45	12.1	11.4	0.429
T stage of cancer											0.266
1	22	(14)	17	(16)	2	(14)	1	(6)	2	(12)	
2	36	(24)	27	(26)	1	(7)	5	(29)	3	(18)	
3	31	(20)	16	(15)	6	(43)	4	(24)	5	(29)	
4	57	(37)	41	(39)	2	(14)	7	(41)	7	(41)	
Surgery Hx	124	(81)	98	(93)	1	(7)	12	(71)	13	(76)	<0.001
Radiotherapy Hx	101	(66)	59	(56)	14	(100)	13	(76)	15	(88)	0.001
Both Surgery and radiotherapy Hx	73	(48)	53	(50)	1	(7)	8	(47)	11	(65)	<0.001
Multiple HNCs	46	(30)	40	(38)	0	(0)	2	(12)	4	(24)	0.006
NG tude dependency	24	(16)	16	(15)	1	(7)	2	(12)	5	(29)	0.069

* Indicated in months. Abbreviations: Dx = diagnosis; HNC = head and neck cancer; Hx = history; IQR = interquartile range; NG = nasogastric; SD = standard deviation.

**Table 2 jcm-11-00692-t002:** Swallowing scores for different head and neck cancers.

			Different Tumor Types:								
Swallowing Scores:	Total *n* = 153		Oral Cavity *n* = 105		NPC *n* = 14		Oropharynx *n* = 17		Hypopharynx *n* = 17		
	Median (IQR)		Median (IQR)		Median (IQR)		Median (IQR)		Median (IQR)		*p*-Value
PAS_L	1	(3)	1	(1)	2.5	(5)	3	(3)	2	(3)	0.010
PAS_SL	1	(3)	1	(1)	2.5	(5)	3	(2)	1	(3)	0.020
PAS_S	1	(3)	1	(3)	2.5	(4)	4	(5)	1	(3)	0.033
BRS_L	3	(3)	3	(3)	4	(2)	4	(1)	2	(3)	0.043
BRS_SL	3	(3)	3	(3)	3.5	(2)	4	(1)	2	(3)	0.211
BRS_S	3	(3)	3	(3)	2.5	(3)	4	(1)	2	(3)	0.505
FOIS *	4.5	(2)	5	(1)	6	(2)	4	(1)	3	(3)	0.014

* For FOIS score, a total of 126 analytical sample were included. This contained 91, 14, 11, 10 subjects from oral cavity, NPC, oropharynx and hypopharynx group, respectively. Abbreviations: BRS = Bolus Residue Scale; FOIS = Functional Oral Intake Scale; IQR = interquartile range; L = liquid; NPC = nasopharyngeal carcinoma; PAS = Penetration Aspiration Scale; SL = semiliquid; S = solid.

**Table 3 jcm-11-00692-t003:** Ordinal logistic regression of VFSS outcomes.

	PAS_L			PAS_SL			PAS_S		
Tumor types	OR	(95% CI)	*p*	OR	(95% CI)	*p*	OR	(95% CI)	*p*
Oral cavity	ref.			ref.			ref.		
NPC	1.17	(0.25, 5.56)	0.847	1.91	(0.39, 9.43)	0.425	1.05	(0.21, 5.21)	0.956
Oropharynx	2.82	(1.03, 7.72)	0.043	3.64	(1.32, 10.03)	0.012	3.17	(1.15, 8.73)	0.025
Hypopharynx	3.00	(1.11, 8.10)	0.030	2.08	(0.75, 5.78)	0.161	1.90	(0.68, 5.32)	0.221
Male	1.46	(0.31, 6.93)	0.635	1.59	(0.32, 7.93)	0.575	0.56	(0.13, 2.35)	0.427
Age	1.04	(1.01, 1.08)	0.022	1.05	(1.01, 1.09)	0.012	1.04	(1.00, 1.08)	0.050
Duration of disease	1.02	(0.87, 1.18)	0.838	1.03	(0.88, 1.20)	0.709	1.04	(0.89, 1.21)	0.661
T stage:									
1	ref.			ref.			ref.		
2	0.65	(0.22, 1.91)	0.435	0.62	(0.20, 1.95)	0.389	0.49	(0.16, 1.49)	0.207
3	0.79	(0.26, 2.39)	0.670	0.62	(0.21, 1.85)	0.414	0.63	(0.20, 1.96)	0.421
4	0.69	(0.25, 1.92)	0.478	0.94	(0.34, 2.60)	0.898	0.88	(0.32, 2.45)	0.812
Surgery Hx	0.85	(0.30, 2.46)	0.768	1.02	(0.34, 3.03)	0.974	0.83	(0.28, 2.46)	0.740
Radiotherapy Hx	1.49	(0.68, 3.28)	0.325	1.31	(0.59, 2.90)	0.501	1.23	(0.55, 2.75)	0.607
Multiple HNCs	0.80	(0.37, 1.73)	0.568	1.23	(0.57, 2.66)	0.605	0.99	(0.45, 2.16)	0.977
	BRS_L			BRS_SL			BRS_S		
Tumor types	OR	(95% CI)	*p*	OR	(95% CI)	*p*	OR	(95% CI)	*p*
Oral cavity	ref.			ref.			ref.		
NPC	1.49	(0.33, 6.71)	0.604	1.38	(0.31, 6.11)	0.672	0.73	(0.16, 3.30)	0.681
Oropharynx	3.12	(1.11, 8.76)	0.031	2.86	(1.04, 7.86)	0.042	3.01	(1.08, 8.39)	0.035
Hypopharynx	0.64	(0.24, 1.71)	0.373	0.72	(0.27, 1.91)	0.504	1.15	(0.43, 3.05)	0.784
Male	1.74	(0.42, 7.13)	0.444	0.94	(0.24, 3.73)	0.935	0.72	(0.18, 2.89)	0.639
Age	1.02	(0.99, 1.05)	0.272	1.00	(0.97, 1.04)	0.792	1.01	(0.98, 1.04)	0.623
Duration of disease	1.07	(0.92, 1.24)	0.367	0.97	(0.84, 1.13)	0.725	1.00	(0.86, 1.16)	0.974
T stage:									
1	ref.			ref.			ref.		
2	0.55	(0.20, 1.48)	0.235	0.42	(0.15, 1.13)	0.085	0.39	(0.14, 1.07)	0.066
3	0.91	(0.32, 2.60)	0.857	0.69	(0.24, 1.95)	0.481	0.85	(0.30, 2.43)	0.757
4	0.82	(0.32, 2.12)	0.680	0.71	(0.27, 1.84)	0.478	0.71	(0.27, 1.85)	0.483
Surgery Hx	0.65	(0.23, 1.86)	0.424	0.98	(0.35, 2.78)	0.971	1.19	(0.42, 3.39)	0.743
Radiotherapy Hx	0.72	(0.36, 1.47)	0.373	0.66	(0.32, 1.33)	0.244	0.60	(0.29, 1.23)	0.162
Multiple HNCs	1.89	(0.94, 3.80)	0.076	2.12	(1.05, 4.27)	0.036	2.28	(1.12, 4.63)	0.023

Abbreviation: BRS = Bolus Residue Scale; Cl = confidence interval; Hx = history; L = liquid; NPC = nasopharyngeal carcinoma; *p* = *p* value; PAS = Penetration Aspiration Scale; ref = reference; SL = semiliquid; S = solid.

**Table 4 jcm-11-00692-t004:** Ordinal logistic regression of FOIS outcomes.

	FOIS		
Tumor types	OR	(95% CI)	*p*
Oral cavity	ref.		
NPC	0.05	(0.00, 0.69)	0.025
Oropharynx	1.25	(0.15, 10.25)	0.833
Hypopharynx	4.58	(0.64, 32.60)	0.129
Male	0.58	(0.06, 5.88)	0.647
Age	1.04	(0.98, 1.09)	0.172
Duration of disease	1.11	(0.89, 1.38)	0.371
T stage:			
1	ref.		
2	0.99	(0.18, 5.58)	0.991
3	0.71	(0.09, 5.39)	0.738
4	1.09	(0.21, 5.60)	0.918
Surgery Hx	0.41	(0.06, 2.74)	0.354
Radiotherapy Hx	5.06	(1.36, 18.87)	0.016
Multiple HNCs	1.72	(0.53, 5.61)	0.371

Abbreviation: Cl = confidence interval; FOIS = Functional Oral Intake Scale; Hx = history; *p* = *p* value; ref = reference.

**Table 5 jcm-11-00692-t005:** Number of patients with abnormal swallowing function among different tumor types.

	Total		Oral Cavity		NPC		Oropharynx		Hypopharynx		
	*n* = 153		*n* = 105		*n* = 14		*n* = 17		*n* = 17		
	*n*	(%)	*n*	(%)	*n*	(%)	*n*	(%)	*n*	(%)	*p*-Value
PAS_L ≥ 3	48	(31)	24	(23)	7	(50)	9	(53)	8	(47)	0.010
PAS_SL ≥ 3	47	(31)	24	(23)	7	(50)	9	(53)	7	(41)	0.016
PAS_S ≥ 3	51	(33)	27	(26)	7	(50)	10	(59)	7	(41)	0.019
BRS_L ≥ 4	63	(41)	41	(39)	8	(57)	9	(53)	5	(29)	0.306
BRS_SL ≥ 4	65	(42)	44	(42)	7	(50)	9	(53)	5	(29)	0.515
BRS_S ≥ 4	66	(43)	44	(42)	6	(43)	9	(53)	7	(41)	0.859

Abbreviation: BRS = Bolus Residue Scale; L = liquid; NPC = nasopharyngeal carcinoma; PAS = Penetration Aspiration Scale; SL = semiliquid; S = solid.

**Table 6 jcm-11-00692-t006:** Odds ratios of abnormal VFSS findings from multivariate logistic regression.

	PAS_L			PAS_SL			PAS_S		
Tumor types	OR	(95% CI)	*p*	OR	(95% CI)	*p*	OR	(95% CI)	*p*
Oral cavity	ref.			ref.			ref.		
NPC	2.53	(0.42, 15.1)	0.308	2.17	(0.37, 12.75)	0.392	0.88	(0.15, 5.15)	0.890
Oropharynx	3.51	(1.1, 11.22)	0.034	3.41	(1.08, 10.75)	0.036	3.49	(1.11, 10.97)	0.032
Hypopharynx	3.06	(0.98, 9.53)	0.054	2.23	(0.71, 6.98)	0.167	1.85	(0.59, 5.78)	0.289
Male	1.38	(0.24, 8.07)	0.722	1.45	(0.25, 8.47)	0.682	0.72	(0.15, 3.59)	0.689
Age	1.06	(1.01, 1.11)	0.018	1.04	(1.00, 1.09)	0.060	1.03	(0.99, 1.07)	0.195
Duration of disease	0.98	(0.82, 1.16)	0.784	1.03	(0.86, 1.22)	0.783	1.05	(0.88, 1.25)	0.567
T stage:									
1	ref.			ref.			ref.		
2	1.26	(0.33, 4.82)	0.734	1.07	(0.28, 4.06)	0.917	0.75	(0.21, 2.72)	0.660
3	1.06	(0.27, 4.22)	0.930	1.15	(0.30, 4.45)	0.844	1.03	(0.28, 3.83)	0.967
4	1.16	(0.32, 4.16)	0.823	1.41	(0.40, 4.91)	0.594	1.14	(0.34, 3.79)	0.830
Surgery Hx	1.03	(0.30, 3.59)	0.960	0.93	(0.27, 3.19)	0.905	0.52	(0.15, 1.77)	0.297
Radiotherapy Hx	1.70	(0.65, 4.44)	0.278	1.62	(0.63, 4.14)	0.319	1.34	(0.53, 3.36)	0.537
Multiple HNCs	0.81	(0.32, 2.06)	0.655	1.06	(0.42, 2.62)	0.909	0.96	(0.39, 2.36)	0.931
	BRS_L			BRS_SL			BRS_S		
Tumor types	OR	(95% CI)	*p*	OR	(95% CI)	*p*	OR	(95% CI)	*p*
Oral cavity	ref.			ref.			ref.		
NPC	1.27	(0.23, 7.09)	0.789	1.08	(0.19, 6.10)	0.927	0.80	(0.13, 4.92)	0.806
Oropharynx	1.83	(0.59, 5.61)	0.294	1.93	(0.63, 5.90)	0.251	2.06	(0.67, 6.39)	0.210
Hypopharynx	0.66	(0.20, 2.16)	0.488	0.62	(0.19, 2.00)	0.420	1.12	(0.36, 3.45)	0.843
Male	2.60	(0.43, 15.66)	0.298	2.31	(0.40, 13.30)	0.348	2.03	(0.34, 12.24)	0.441
Age	1.03	(0.99, 1.07)	0.215	1.02	(0.98, 1.06)	0.313	1.03	(0.99, 1.07)	0.130
Duration of disease	1.08	(0.92, 1.27)	0.366	0.97	(0.82, 1.14)	0.687	0.96	(0.81, 1.15)	0.673
T stage:									
1	ref.			ref.			ref.		
2	0.70	(0.22, 2.21)	0.539	0.54	(0.17, 1.69)	0.290	0.50	(0.16, 1.61)	0.248
3	1.44	(0.43, 4.84)	0.553	1.05	(0.32, 3.43)	0.940	1.27	(0.38, 4.22)	0.702
4	1.25	(0.42, 3.74)	0.694	1.04	(0.35, 3.05)	0.950	1.02	(0.34, 3.08)	0.967
Surgery Hx	0.53	(0.16, 1.75)	0.296	0.72	(0.22, 2.38)	0.584	0.78	(0.23, 2.63)	0.691
Radiotherapy Hx	0.69	(0.30, 1.58)	0.384	0.73	(0.33, 1.65)	0.453	0.65	(0.29, 1.48)	0.309
Multiple HNCs	1.95	(0.87, 4.37)	0.105	1.96	(0.89, 4.34)	0.097	2.42	(1.08, 5.42)	0.031

Abbreviation: BRS = Bolus Residue Scale; Cl = confidence interval; Hx = history; L = liquid; NPC = nasopharyngeal carcinoma; *p* = *p* value; PAS = Penetration Aspiration Scale; ref = reference; SL = semiliquid; S = solid.

## Data Availability

The data are not publicly available due to data privacy of individual patients.

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
