# Peer review of "Evaluation of Objective and Subjective Swallowing Outcomes in Patients with Dysphagia Treated for Head and Neck Cancer"

_jcm, 2022, doi:10.3390/jcm11030692_

Round 1

Reviewer 1 Report

The authors present an interesting and relevant manuscript which is well written. Some aspects need to be considered before publication:

-Table 1: No information for the abbreviation "Dx" is given. I suppose it means diagnosis )?). However, the table still remains unclear. What diagnosis? Of the tumor? how can the diagnosis be shorter than surgery? Of dysphagia? How can you have the diagnosis if you perform VFS only at the day of study? Maybe symptoms? please clarify

-English "this groups' patients' odds ...." line 231 

-Paragraph "tumor types and swallowing outcomes": The authors somewhatinterchangeably use "risks" and "odds". “Risk” refers to the probability of occurrence of an event or outcome. Statistically, risk = chance of the outcome of interest/all possible outcomes. The term “odds” is often used instead of risk. “Odds” refers to the probability of occurrence of an event/probability of the event not occurring. At first glance, though these two concepts seem similar and interchangeable, there are important differences that dictate where the use of either of these is appropriate. Further, table 2 suggests, that the authors actually calculated odds RATIOS, as they state from line 234 following. I suggest, that the authors carefully revise their manuscript and use the appropriate wording.

-Table 2 a/b/c. It is quite uncommon to use table "panels" as done by the authors. I suggest they either break the table apart or use additional data in a supplementary table.

-While the authors used p-values of =.xxx in the majority of the manuscript, they use =.xxxx in table 3A. Please revise this table to fit the rest of the manuscript

-Please shorten the conclusions

Author Response

The authors present an interesting and relevant manuscript which is well written. Some aspects need to be considered before publication:

-Table 1: No information for the abbreviation "Dx" is given. I suppose it means diagnosis )?). However, the table still remains unclear. What diagnosis? Of the tumor? how can the diagnosis be shorter than surgery? Of dysphagia? How can you have the diagnosis if you perform VFS only at the day of study? Maybe symptoms? please clarify

Thank you for the constructive comments. We apologize for not making it clear. Dx stands for diagnosis, and in this study it referred to diagnosis of head and neck cancer. We have revised the column name added the abbreviations into the footnote. And of duration of the diagnosis was originally presented in years while the duration from surgery was presented in months. That was the reason why the values were lower in the duration of diagnosis. To avoid confusion, we have unified all duration values in months. Please refer to the adjusted Table 1. Thank you.

-English "this groups' patients' odds ...." line 231 

Thank you. We have rewritten the sentence into “It was noted that in all types of swallowing, oropharynx group manifested consistent ORs at approximately 3 when compared to the oral cavity group.” (Line 233-235 )

-Paragraph "tumor types and swallowing outcomes": The authors somewhat interchangeably use "risks" and "odds". “Risk” refers to the probability of occurrence of an event or outcome. Statistically, risk = chance of the outcome of interest/all possible outcomes. The term “odds” is often used instead of risk. “Odds” refers to the probability of occurrence of an event/probability of the event not occurring. At first glance, though these two concepts seem similar and interchangeable, there are important differences that dictate where the use of either of these is appropriate. Further, table 2 suggests, that the authors actually calculated odds RATIOS, as they state from line 234 following. I suggest, that the authors carefully revise their manuscript and use the appropriate wording.

Thank you for reminding us of the difference between ‘risk’ and ‘odds’. We have gone through the manuscript again and revised all terms with appropriate wordings.

-Table 2 a/b/c. It is quite uncommon to use table "panels" as done by the authors. I suggest they either break the table apart or use additional data in a supplementary table.

Thank you for the advice. To make it clearer and simpler, we have modified all tables into separate numbering rather than in panels.

-While the authors used p-values of =.xxx in the majority of the manuscript, they use =.xxxx in table 3A. Please revise this table to fit the rest of the manuscript

Thank you for pointing this out. We have revised it accordingly and unified it as p=.xxx.

-Please shorten the conclusions

Thank you for the advice. We have shortened the conclusion and please refer to Line 348-357: “By using objective assessment tool, we found that the risk of aspiration and having bolus residue was high for patients with oropharyngeal tumors. In contrast, patients with nasopharyngeal tumors revealed better swallowing function using subjective assessment. These findings indicated that compared to exclusively using only a single tool, complementary application of both objective and subjective assessments may improve evaluations of dysphagia in patients with HNC. We also found that increased age, having multiple HNCs, and the receipt of radiotherapy were predictive factors for poor PAS scores, BRS scores, and FOIS scores, respectively. Clinicians should take these risk factors into account when they assess swallowing function and design rehabilitation programs for these patients.”

Reviewer 2 Report

The article entitled “Evaluation of objective and subjective swallowing outcomes in patients with dysphagia treated for different types of head and neck cancer” analyzed objective and subjective swallowing analyses in 153 HNC patients with dysphagia between 2016 and 2021.

The followings are my comments:

  1. This study investigated an issue of clinical significance. Early identification of patients who may have swallowing disorders facilitates early intervention in patients and improves their nutritional function and quality of life, authors can enhance the description in the Discussion section about the value of early identification.
  2. From the title and abstract, it seems that the swallowing assessments were performed only in patients with dysphagia; however, in method section, seems to be “extensively performed (Line 133)”. Please give description about patient selection, is FOIS score the screening tool? If yes, it could be more relevant to the description “This indicates using only subjective assessments may not allow for accurate evaluations of swallowing function in patients treated for HNC.”
  3. In Line 32-34 the author mentioned p16 positive oropharyngeal cancer, the staging method also is AJCC 8th The readers will look forward to the author's analysis on p16 stratification. In addition, most of the oropharyngeal cancer patient in this study received surgery, is that related to the p16 +/- distribution?
  4. The exclusion criteria are not clear enough, only 4 patients had dysphagia prior to HNC treatment, they should be all described (not only stroke). The Table 1 should be remade.
  5. In Line 83-84 “the patient had already received operative treatment and/or radiotherapy” seems all the patient received surgery, and obviously wrong. The description of the treatment modality in the article is too brief, whether there is a free flap or other reconstruction for oral/oropharyngeal/hypopharyngeal cancer, and whether partial laryngectomy or TORS is used for hypopharyngeal cancer (total laryngectomy cases were already excluded, it means hypopharyngeal cancer patients with the most severe swallowing impairments are ruled out).
  6. Some critical information is missing. (1) No description about cancer recurrence or residual in this study. (2) The NG tube dependency rate? (3) When the patients start their swallowing rehabilitation (“The median duration between the time the patients underwent surgery to the time they underwent VFSSs was 9.75 months”? (4) The description about multiple cancers.
  7. Why is nasopharyngeal cancer subjectively better but objectively worse swallowing performance than oral-cavity cancer? I cannot find a good explanation in the description section.

Author Response

  1. This study investigated an issue of clinical significance. Early identification of patients who may have swallowing disorders facilitates early intervention in patients and improves their nutritional function and quality of life, authors can enhance the description in the Discussion section about the value of early identification.

Thank you for the valuable comment. We have revised the last paragraph in discussion regarding the clinical significance of early identification.

(Line 349-355) “A combined use of multiple measures may help healthcare professionals to conduct the evaluation of the patients’ swallowing function more comprehensively, therefore aiding in early identification of dysphagia. Dysphagia may result in aspiration, penetration, dehydration, malnutrition, and increase other morbidities in patients with HNC. These would further decrease patients’ nutritional function and quality of life. Our study implied the clinical significance of early identification of the dysphagia, which would facilitate early intervention or management to prevent further complications.”

  1. From the title and abstract, it seems that the swallowing assessments were performed only in patients with dysphagia; however, in method section, seems to be “extensively performed (Line 133)”. Please give description about patient selection, is FOIS score the screening tool? If yes, it could be more relevant to the description “This indicates using only subjective assessments may not allow for accurate evaluations of swallowing function in patients treated for HNC.”

Thank you for the comment. We apologize for the confusion. The swallowing assessment of VFSS/FOIS was performed in patients who complained of difficulty swallowing. Therefore, VFSS was not extensively performed in our hospital. In addition, we did not screen the patients using FOIS. Instead, VFSS and FOIS were performed for these patients after they presented with dysphagia and came to the clinic. Correspondingly, the revised text can be found in:

- (Line 134-135) “To all included patients, VFSS was performed by a speech pathologist with more than 10 years of experience.”

- (Line 78-81) “In this retrospective, cross-sectional, observational study, patients treated for HNC, who had visited the otorhinolaryngology outpatient clinic of the National Cheng Kung University Hospital (NCKUH) with a complaint of difficulty swallowing and undergoing VFSSs between January 2016 and March 2021, were included.”

  1. In Line 32-34 the author mentioned p16 positive oropharyngeal cancer, the staging method also is AJCC 8th The readers will look forward to the author's analysis on p16 stratification. In addition, most of the oropharyngeal cancer patient in this study received surgery, is that related to the p16 +/- distribution?

Thank you for raising this comment. We mentioned HPV-related tumors in the introduction with only the intention to emphasize how advancement in clinical care had increased the survival of patients with HNC. It is recommended in the guideline that p16 exam would help in deciding treatment modality for patients with oropharynx tumor for potentially better prognosis. This implies that p16 could be represented as a proxy of treatment modality, which affects outcomes of interest. However, since we were already concerned about treatment modality and had adjusted it in the regression models, there would be no need to adjust p16 status additionally, since the control of this factor would result in potential over-adjustment. However, to avoid confusion, we have revised the text into the following (Line 32-34): “Survival of patients with head and neck cancer (HNC) has improved considerably in recent decades due to improved public health awareness and the use of management strategies, such as radiotherapy or surgery.”

  1. The exclusion criteria are not clear enough, only 4 patients had dysphagia prior to HNC treatment, they should be all described (not only stroke). The Table 1 should be remade.

Thank you for the comment. The 4 patients were excluded all because of stroke, and we have recreated Figure 1 to make it clearer. We also revised the corresponding text concerning the exclusion criteria in methods.

(Line 88-89) “Patients with underlying conditions, such as stroke, dementia, or neuromuscular diseases that had already caused or may cause dysphagia were excluded.”

  1. In Line 83-84 “the patient had already received operative treatment and/or radiotherapy” seems all the patient received surgery, and obviously wrong. The description of the treatment modality in the article is too brief, whether there is a free flap or other reconstruction for oral/oropharyngeal/hypopharyngeal cancer, and whether partial laryngectomy or TORS is used for hypopharyngeal cancer (total laryngectomy cases were already excluded, it means hypopharyngeal cancer patients with the most severe swallowing impairments are ruled out).

Thank you for the advice. We have revised the text accordingly.

(Line 81-86) “A patient was included in the study population if they had been diagnosed with HNC, had already received operative treatment, radiotherapy or both, and with VFSS results of appropriate quality, including accurate observations of pharyngeal structures. The operative treatments in these patients with HNC including partial laryngectomy, glossectomy, pharyngectomy and local or free flap reconstruction.”

  1. Some critical information is missing. (1) No description about cancer recurrence or residual in this study. (2) The NG tube dependency rate? (3) When the patients start their swallowing rehabilitation (“The median duration between the time the patients underwent surgery to the time they underwent VFSSs was 9.75 months”? (4) The description about multiple cancers.

Thank you for the advice.

(1) & (4)

For the variable of multiple cancers, we referred to HNC recurrence or other primary HNC at different sites. For example, if an included patient with tongue cancer had recurrence or had been diagnosed with buccal cancer after, the patient would be considered having multiple HNCs. We have revised the description in corresponded paragraph. In addition, we unified ‘multiple cancers’ into ‘multiple HNCs’ for better understanding.

(Line 155-157) “we determined whether the patients had been diagnosed with multiple HNCs, including recurrent cancer of the original or a second primary HNC at a different site.”

(2)

NG tube dependency was already considered in the FOIS scale when evaluating subjective swallowing functions. We added the NG tube dependency rate in Table 1 as well. Thank you.

(3)

For each patient, we started swallowing rehabilitation after establishment of dysphagia diagnosis by VFSS, which was performed at the time of complaint of difficulty swallowing by the patient with their conditions stabilized after completing HNC treatment, including surgery or radiotherapy regimen. This duration varied a lot in different patients, as shown by the wide standard deviation reported in Table 1. Through this study, we also acknowledged that the time between HNC treatment and VFSS exam was quite long, and this prompted us to reflect on our clinical practice that we could evaluate the swallowing functions of these patients more actively.

  1. Why is nasopharyngeal cancer subjectively better but objectively worse swallowing performance than oral-cavity cancer? I cannot find a good explanation in the description section.

Thank you for the comment. In this study regarding the outcomes from nasopharynx group, the results had only indicated better subjective outcomes but not worse objective outcome (From the main results, it seemed that odds of NPC group were sometimes greater than 1, however, they were not significant since the sample size is not very large and the variation of risk estimates became wider.). We have explained discussed the potential reason of better subjective swallowing function in NPC group in discussion section (Line 325-331).

Round 2

Reviewer 2 Report

The author has made comprehensive revisions as suggested, and the article has been significantly improved and has reached an acceptable level for the present form.

Author Response

Thank you again for your constructive comments/suggestions. We really appreciated it!